# Topological current divider in a Chern insulator junction

**Dmitry Ovchinnikov** [1,7], **Jiaqi Cai** [1,7], **Zhong Lin**[1], **Zaiyao Fei** [1], **Zhaoyu Liu** [1], **Yong-Tao Cui** [2], **David H. Cobden** [1], **Jiun-Haw Chu** [1], **Cui-Zu Chang** [3], **Di Xiao** [1,4,5], **Jiaqiang Yan**[6] **& Xiaodong Xu** [1,4] ✉

A Chern insulator is a two-dimensional material that hosts chiral edge states produced by the combination of topology with time reversal symmetry breaking. Such edge states are perfect one-dimensional conductors, which may exist not only on sample edges, but on any boundary between two materials with distinct topological invariants (or Chern numbers). Engineering of such interfaces is highly desirable due to emerging opportunities of using topological edge states for energy-efficient information transmission. Here, we report a chiral edge-current divider based on Chern insulator junctions formed within the layered topological magnet $MnBi_2Te_4$. We find that in a device containing a boundary between regions of different thickness, topological domains with different Chern numbers can coexist. At the domain boundary, a Chern insulator junction forms, where we identify a chiral edge mode along the junction interface. We use this to construct topological circuits in which the chiral edge current can be split, rerouted, or switched off by controlling the Chern numbers of the individual domains. Our results demonstrate $MnBi_2Te_4$ as an emerging platform for topological circuits design.

Quantum matter with non-trivial topologies of electronic bands holds great potential in advancing next-generation information, computing, and storage technologies[1-3]. Surface and edge modes in gapped topological systems[1] are exciting venues for exploring how information can be transmitted with minimal dissipation[4,5] and in a non-reciprocal way[6,7]. For example, two-dimensional magnetic topological insulators (also known as Chern insulators[8,9]) possess one-dimensional chiral edge states, in which electrons travel strictly in only one direction and backscattering is topologically forbidden, as in the quantum Hall effect. The direction of propagation of such edge state is controlled by magnetization of the material. The number of chiral edge states is equal to the Chern number $C$ and determines the quantized Hall resistance, $R_{yx} = h/Ce^2$ (ref. 10), where $h$ is Planck's constant and $e$ is the elementary charge.

When two materials with different Chern numbers are put in contact with each other, due to bulk-boundary correspondence chiral edge states can emerge at the interface[1]. This was demonstrated in experiments on two-dimensional electron gas[11] and graphene[12-14] where regions with different $C$ can be created by means of locally changing carrier densities in quantum Hall regime. The Chern insulator networks[15-17] comprised of domains with distinct Chern numbers may lead to complex device architectures, where $C$ of individual domains will control topological current properties, such as current amplitude and propagation direction. Previous attempts to construct such networks which went beyond quantum Hall state relied on chiral edge states existing at domain walls in magnetic topological insulators, since domains of opposite magnetization have opposite Chern number (ref. 18–20). One-dimensional edge modes can also exist on the

---

[1]Department of Physics, University of Washington, Seattle, WA 98195, USA. [2]Department of Physics and Astronomy, University of California, Riverside, CA 92521, USA. [3]Department of Physics, The Pennsylvania State University, University Park, PA 16802, USA. [4]Department of Materials Science and Engineering, University of Washington, Seattle, WA 98195, USA. [5]Pacific Northwest National Laboratory, Richland, WA, USA. [6]Materials Science and Technology Division, Oak Ridge National Laboratory, Oak Ridge, TN 37831, USA. [7]These authors contributed equally: Dmitry Ovchinnikov, Jiaqi Cai. ✉e-mail: xuxd@uw.edu

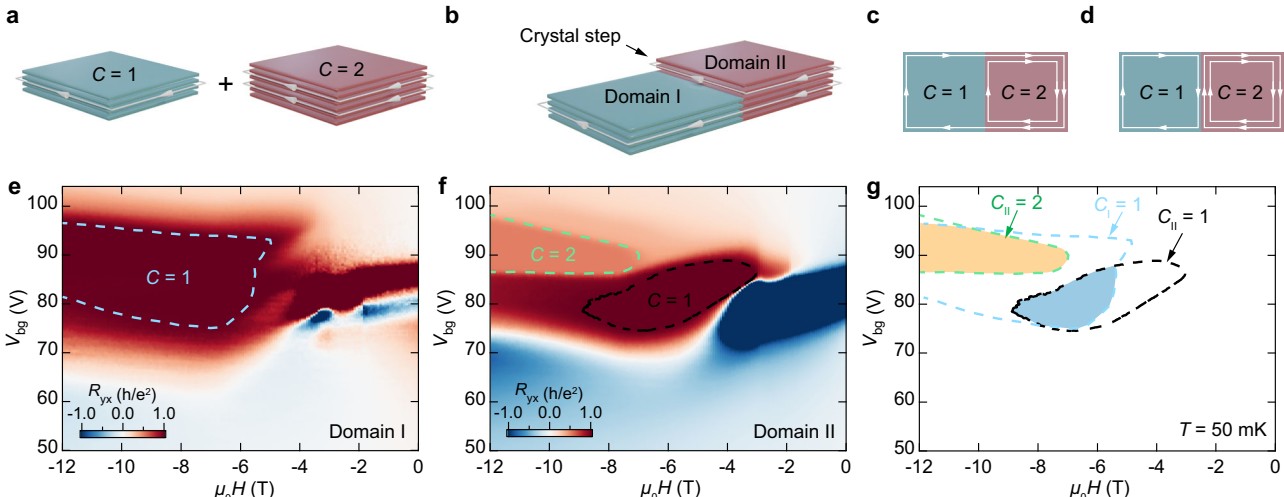

**Fig. 1 | Creating a Chern junction in MnBi$_2$Te$_4$. a–c** Concept of the junction. **a** A thinner MBT flake (Domain I) hosts a chiral edge state with Chern number $C = 1$ while a thicker flake (Domain II) hosts two chiral edge conduction channels with $C = 2$. **b** A Chern junction is formed at a step in thickness in a single flake. **c–d** Show two possible scenarios for the chiral edge states at the junction. **e–g** Characterization of Device 1, which contains a 4-layer (Domain I) to 6-layer (Domain II) junction. All data are taken at $T = 50$ mK. **e** Hall resistance $R_{yx}$ as a function of back gate voltage $V_{bg}$ and magnetic field $\mu_0 H$ for Domain I. The blue dashed line is the contour $R_{yx} = 0.98 \, h/e^2$ and surrounds the parameter region where $C_I = 1$. **f** Similar measurements for Domain II. The black line is at $R_{yx} = 0.98 \, h/e^2$ and indicates the parameter region where $C_{II} = 1$ is located, while the green dashed line is at $R_{yx} = 0.49 \, h/e^2$ and indicates where $C_{II} = 2$ is located. **g** Overlaid contours from (**e**,**f**). The parameter region where the Chern junction exists ($C_I = 1$, $C_{II} = 2$) is shaded orange, and the region where no junction is expected ($C_I = 1$, $C_{II} = 1$) is shaded blue.

crystal step edges of topological materials[21–26], but domain-wall and step-edge states are nearly impossible to harness in an electronic device. As a result[15–17,27], achieving robust and tunable topological circuit elements, such as a topological current divider[27], has remained a challenge.

In this work, we intentionally create a "Chern junction" in MnBi$_2$Te$_4$ (MBT) between domains with $C = 1$ and $C = 2$ and exploit it to demonstrate the basic operations of a topological circuit, including splitting, redirection, and switching of chiral edge currents. MBT is a van der Waals topological antiferromagnet[28] in which each covalently bonded layer (comprising seven atomic planes in the sequence Te-Bi-Te-Mn-Te-Bi-Te) is ferromagnetic with out-of-plane magnetization, while the coupling across the van der Waals gap between layers is antiferromagnetic (AFM)[28]. In few-layer MBT the intertwined magnetic and Chern insulator states are tuneable[29–35] by a combination of factors: magnetic field, which modifies the magnetic state[29,35,36]; electrostatic gating, which tunes the chemical potential relative to the exchange gap[9,29]; and thickness[32]. The corresponding chiral edge states also persist to relatively high temperatures (up to $T \approx 30$ K, ref. 29, 32), another helpful feature for prototyping topological circuits.

## Results
### Chern insulator junction formation
Our Chern junction design is shown in Fig. 1a, b. In addition to the $C = 1$ state in a thin flake[29–32] (labeled Domain I), MBT can host higher Chern number states in a thicker flake[30,32,35] (Domain II). Chern number $C = 2$ state in MnBi$_2$Te$_4$ was recently identified as a combination of $C = 1$, which originates from non-trivial band topology and $\nu = 1$ quantum Hall state[35]. Within our experimental parameter space[32,35], we observe $C = 2$ only in flakes which are $\geq 6$ layers thick (see Methods). In flakes with thickness 4 and 5 layers only $C = 1$ state appears. As a result, in a single flake containing a step in thickness, the thinner (Domain I) and thicker (Domain II) parts can simultaneously be tuned to have $C = 1$ and $C = 2$, respectively, creating a Chern junction at the boundary (Fig. 1b). There are two possible scenarios for the chiral edge states at the Chern junction: either one of them travels along the boundary and one crosses it (Fig. 1c), or three travel along the boundary and none crosses it (Fig. 1d). We intentionally selected exfoliated MBT flakes with

a 1–2 layer step edge because self-standing 1–2 layer flakes have no edge states, and thus any edge states at the boundary must be associated with the Chern junction.

We focus here on a device (Device 1) which has a 4-layer part (Domain I) and a 6-layer part (Domain II) (Supplementary Fig. 1). We first establish the phase diagrams of the two domains as a function of magnetic field $\mu_0 H$ and back gate voltage $V_{bg}$ by measuring $R_{yx}$ to determine their respective Chern numbers, $C_I$ and $C_{II}$ (see Supplementary Note and Methods). Figure 1e is a 2D color map of $R_{yx}$ as a function of $\mu_0 H$ and $V_{bg}$ at $T = 50$ mK (the longitudinal resistance $R_{xx}$ is shown in Supplementary Fig. 2). Within the blue dashed contour $0.98 \, h/e^2 \leq R_{yx} \leq 1.002 \, h/e^2$, implying $C_I = 1$. Figure 1f is a similar plot for Domain II. Here, within the black dashed contour $0.98 \, h/e^2 \leq R_{yx} \leq 1.002 \, h/e^2$, implying $C_{II} = 1$, and within the green dashed contour $0.49 \, h/e^2 \leq R_{yx} \leq 0.51 \, h/e^2$ implying $C_{II} = 2$ (ref. 35). Figure 1g shows an overlay of these parameter regions for both domains. The region where $C_I = 1$ and $C_{II} = 2$, i.e., where a Chern junction forms, is shaded orange. The Chern junction can be reconfigured (eliminated) by changing the parameters to lie in the blue shaded region where $C_I = C_{II} = 1$.

### Topological current divider
We next establish the existence of chiral edge states at the junction interface. We can probe the configuration of chiral edge states using various contact configurations. To start with, we inject a bias current $I$ (2 nA) at contact 1 and measure the currents flowing to ground through contacts 8 ($I_8$) and 9 ($I_9 = I - I_8$) with all other contact floating. The magnetic field is first set to $\mu_0 H = -12$ T, such that the edge states propagate clockwise around the sample (Fig. 2a). The resulting variation of $I_8$ and $I_9$ with $V_{bg}$ is shown in Fig. 2b. When $V_{bg}$ is well outside the range for which $C_I = 1$ and $C_{II} = 2$, which is roughly $85 \, \text{V} < V_{bg} < 95 \, \text{V}$ (yellow shaded area in Fig. 2b), $I_9$ is several times larger than $I_8$. This is naturally explained by the presence of bulk conductivity (when $V_{bg} < 80$ V or $V_{bg} > 95$ V), the path through the bulk from 1 to 9 being shorter than that from 1 to 8. In contrast, when $85 \, \text{V} < V_{bg} < 95 \, \text{V}$, where $C_I = 1$ and $C_{II} = 2$, we observe $I_9 = I_8$. This measurement unambiguously identifies that Fig. 1c depicts the correct interfacial chiral edge conduction channel configuration. In case the physical picture depicted in

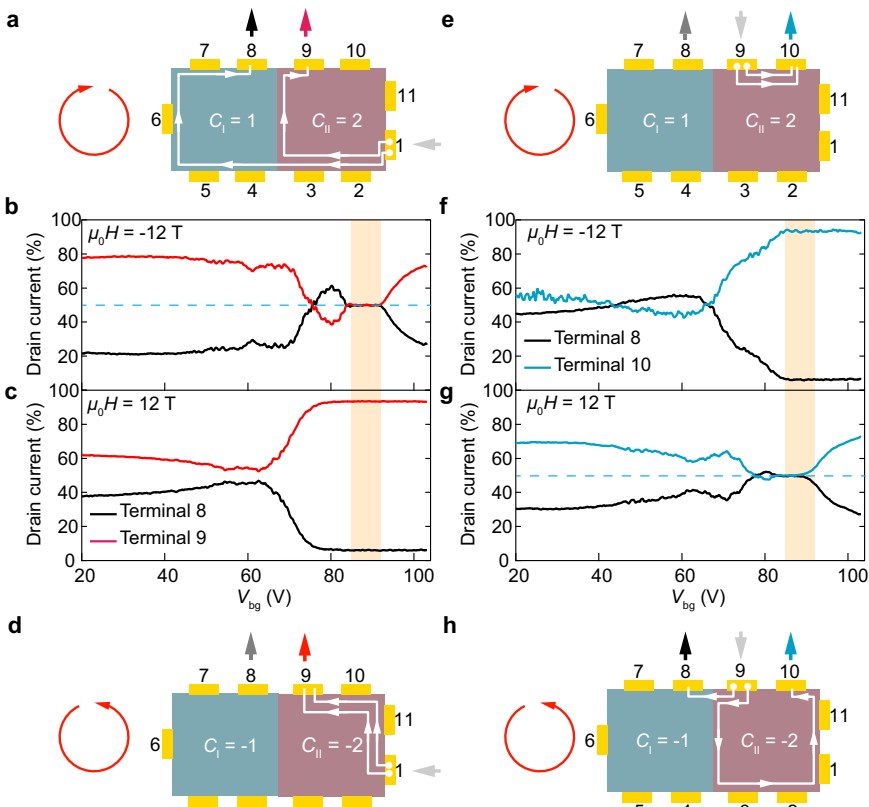

**Fig. 2 | Detection of chiral edge states at the Chern junction. a–d** A bias current of 2 nA is injected at contact 1 (source) and the drain current is measured at contacts 8 (black) and 9 (red) with other contacts floating. **a** Schematics for current flow through edge states at $\mu_0H = -12$ T when $C_I = 1$ and $C_{II} = 2$. **b** Drain current normalized to the total current as a function of back gate voltage $V_{bg}$ for contacts 8 and 9. Yellow shaded area denotes the gate range where Chern insulator junction forms and functions as a 1:1 current divider. **c** Same measurement as shown in (**b**) but performed at $\mu_0H = 12$ T, and (**d**) is the corresponding edge current flow

diagram. **e–h** Similar as (**a–d**) but with the bias current injected at contact 9 (source) and the drain current measured at contacts 8 (black) and 10 (blue) with other contacts floating. **e** Schematics for current flow through edge states at $\mu_0H = -12$ T when $C_I = 1$ and $C_{II} = 2$. **f** Drain current normalized to the total current versus $V_{bg}$ for contacts 8 and 10. **g** Same measurement as shown in (**f**) but at $\mu_0H = 12$ T, and (**h**) is the corresponding edge current flow schematics. Red circular arrow denotes the chirality of the edge state, which is clockwise for negative magnetic field (**a** and **e**) and counterclockwise for positive field (**d** and **h**).

Fig. 1d would be a valid one, all current should flow from contact 1 to 9 and give vanishing $I_8$. The configuration in Fig. 1c leads to the current flow pattern sketched using white lines in Fig. 2a, which leads to $I_9 = I_8$ (see Methods and Supplementary Fig. 3 for measurement details).

When the magnetic field is reversed to +12 T (Fig. 2c), the behavior is quite different and now $I_9$ dominates $I_8$ when 85 V < $V_{bg}$ < 95 V. This is because the sample magnetization is now reversed, switching the Chern numbers to $C_I = -1$ and $C_{II} = -2$ and causing the chiral edge states to reverse direction so that they now both directly convey current from contact 1 to contact 9, as sketched in Fig. 2d. Using other contact configurations we can confirm that this picture is correct. For example, when we inject the current at contact 9 and measure the currents to ground through contacts 8 and 10 at $\mu_0H = -12$ T, when 85 V < $V_{bg}$ < 95 V most of the current is delivered directly to contact 10 via both chiral edge states (Fig. 2e, f). However, when the field is reversed to +12 T, equal currents flow out of contacts 8 and 9 (Fig. 2g) because the junction then acts as a 1:1 current divider (Fig. 2h). Results for a third configuration are shown in Supplementary Fig. 4.

The above results demonstrate that a chiral edge state can exist at a crystalline step edge in MBT. Unlike the topological edge states that have been detected in other systems using scanning probe techniques[21–26], this chiral edge state can be switched, detected by transport, and harnessed for multi-terminal devices. The results presented in Fig. 2 demonstrate that the Chern junction can function as a simple circuit element: a current signal can be either split equally between two outputs or routed to a single output by flipping the

direction of the chiral edge mode. The existence of chiral edge states in this device makes the use of different contacts within a single topological domain equivalent[6,7]. To demonstrate this explicitly, in Fig. 3a we show that in the appropriate gate range (85 V < $V_{bg}$ < 95 V), the current splitting obtained using any of contacts 4, 5, 6 or 7 to Domain I is identical to that using contact 8 as in Fig. 2a, b, i.e., the current injected at contact 1 in Domain II is always divided 1:1 at the Chern junction (see also Supplementary Fig. 5). Note that small changes in the gate range for current divider operation in Fig. 3a might be related to small Fermi level variations across the device, which has a large lateral size of ~100 μm.

### Control of Chern junction

The gate voltage and magnetic field provide additional control of the Chern junction by altering $C_I$ and $C_{II}$. Figure 3c shows the division of current injected at contact 1 between contacts 8 and 9 as a function of $V_{bg}$ and $\mu_0H$. The black dashed line outlines the region where $C_I = 1$, $C_{II} = 2$ and the device works as a 1:1 chiral edge current divider, as depicted in Fig. 3b. The white dashed line outlines the region with $C_I = C_{II} = 1$, where there is no edge state along the boundary and the current is not divided but is all conveyed to contact 8 via a single edge state on the perimeter of the flake (Fig. 3d). In the low-field limit ($|\mu_0H| < 4$ T) and close to charge neutrality (85 V < $V_{bg}$ < 95 V), most of the current flows from contact 1 to the nearest contact 9 via bulk conduction, because the chiral edge states do not exist in the AFM phase[29,31,32,35].

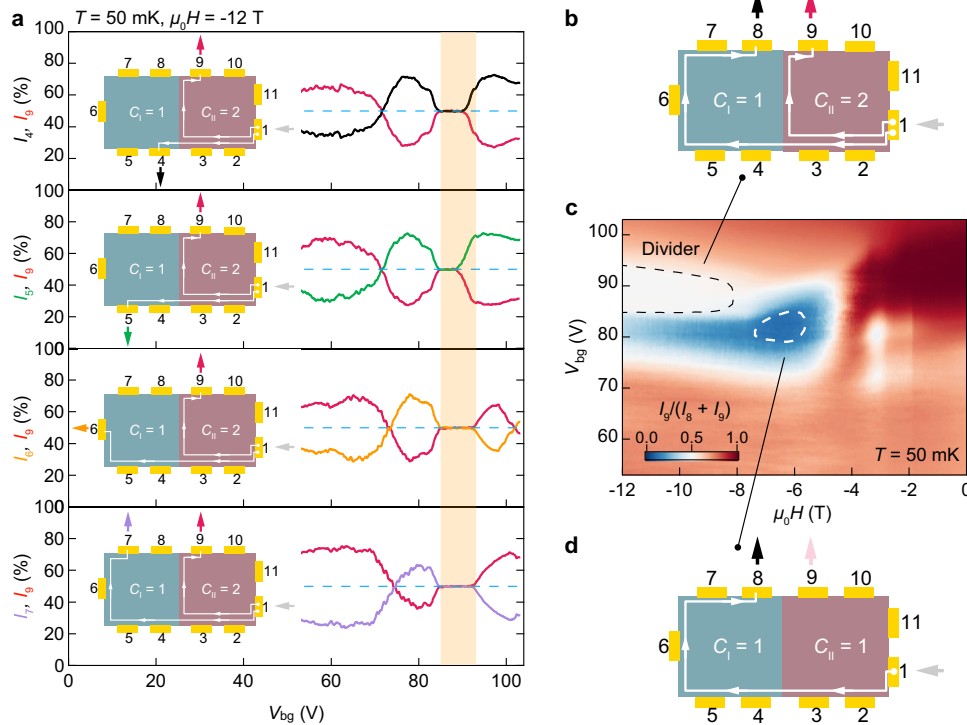

**Fig. 3 | Characterization and control of topological circuit. a** Operation of the dissipationless chiral edge current divider with the bias applied at contact 1 and various drain configurations at $\mu_0 H = -12$ T. Insets are the current flow diagrams. The semi-transparent yellow bar highlights the gate voltage range for a functional current divider, where the injected current splits equally at the Chern junction. The nearly identical outputs with different drain configurations demonstrate the dissipationless nature of the chiral edge current divider. **b–d** Controllable operation of the chiral edge current divider using gate voltage and magnetic fields. Bias is applied at contact 1 and drain at 8 and 9. **c** 2D color map of $I_9/(I_9 + I_8)$ versus $\mu_0 H$ and

$V_{bg}$. Black dashed line outlines region where Domains I and II are tuned into $C_I = 1$ and $C_{II} = 2$, respectively. In this phase space, the injected current is split equally into two chiral edge conduction channels at the junction, i.e., $I_9/(I_9 + I_8) = 50\%$. The current flow is indicated in (**b**). By tuning the gate voltage and magnetic field, both Domain I and II can be set to $C_I = C_{II} = 1$, outlined by white dashed line in the middle panel. In this phase space, the divider is switched off. As shown in (**d**) all current injected at contact 1 flows to contact 8 through single chiral edge channel and does not reach contact 9, i.e., $I_9/(I_9 + I_8)$ nearly vanishes.

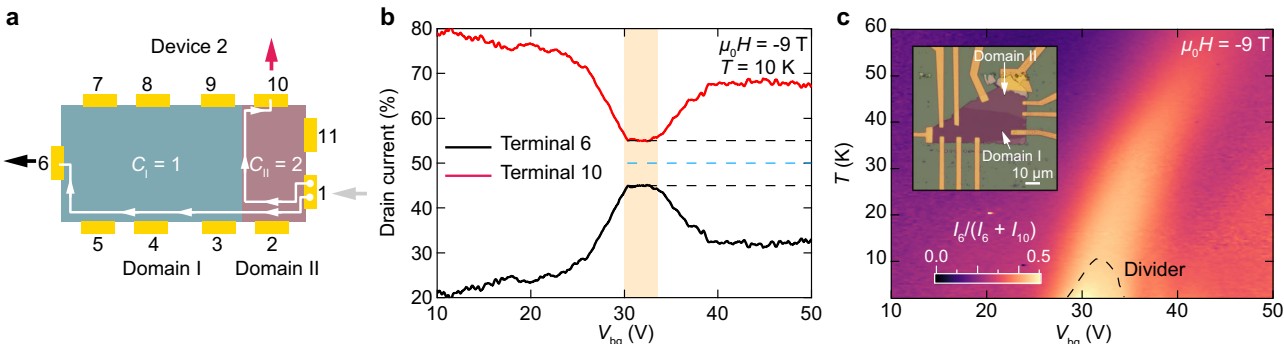

**Fig. 4 | Robustness of the chiral edge current divider to temperature. a** Device 2 measurement scheme. Domains I and II are tuned to Chern numbers of 1 and 2, respectively. Current is injected at contact 1 (gray arrow) and detected at contacts 6 (black) and 10 (red). Winding of edge states is clockwise at $\mu_0 H = -9$ T. Yellow shaded area denotes the gate range where Chern insulator junction forms and functions as a 1:1 current divider. **b** Normalized drain current versus $V_{bg}$ for contacts

6 and 10 at $\mu_0 H = -9$ T, $T = 10$ K. Blue line denotes 50%, corresponding to perfect current division at the Chern insulator junction. Black lines denote 45% and 55%. **c** 2D color map of normalized current from contact 1 to contact 6: $I_6/(I_6 + I_{10})$. Dashed line corresponds to 5% deviation from equal current division. Inset−optical micrograph of Device 2.

An asset of MBT devices is the persistence of the chiral edge states to elevated temperature[29,32]. Figure 4 shows the efficiency of topological current division as a function of temperature in another device (Device 2) in which Domains I and II differ in thickness by only 1 layer (Supplementary Fig. 6). Figure 4a shows the divider configuration (similar to that shown in Fig. 2a), and Fig. 4b shows how the current is actually divided between contacts 6 ($I_6$) and 10

($I_{10}$) at $T = 10$ K. For $30$ V $< V_{bg} < 33$ V (shaded yellow region) and $\mu_0 H = -9$ T, when the required Chern junction condition ($C_I = 1$, $C_{II} = 2$) is met, the deviation from ideal 1:1 current division (blue dashed line) is <5% (black dashed lines). The evolution of $I_6$ with temperature and gate voltage is plotted in Fig. 4c. Even at 30 K, the deviation under optimal conditions is only 10% (Supplementary Fig. 7).

## Discussion

In conclusion, we have shown the creation and manipulation of chiral edge states at crystal step edges in the topological magnet $MnBi_2Te_4$ and used them to demonstrate proof-of-concept topological circuit elements which can divide and redirect dissipationless current. Both $C = 1$ (ref. [29], [32]) and $C = 2$ (ref. [32]) states persist to high temperatures, which is important for practical applications. In the future, additional degrees of freedom such as multiple top gates can be introduced for further control of Chern numbers locally, while recent developments in controllable molecular beam epitaxy of MBT (ref. [37–40]) offer scalability. Finally, although our results are in the low-frequency limit, the device architecture offers a promising route to higher frequency non-reciprocal switches[6,7] based on van der Waals topological magnets.

## Methods

### Device fabrication

Bulk crystals of $MnBi_2Te_4$ were grown out of a Bi-Te flux as previously reported[41]. Scotch tape exfoliation was used to obtain flakes with thicknesses between 4 and 8 layers on 285 nm $SiO_2$ grown on degenerately p-doped Si wafer. The thickness was determined by combination of optical contrast, RMCD measurements and atomic force microscopy[29]. We identified suitable stepped flakes and cleaned away surrounding bulk flakes with a sharp needle. The flakes were incorporated into back-gated devices by electron beam lithography using polymethyl methacrylate (PMMA) resist, thermal evaporation of Cr (5 nm) and Au (60 nm) and lift off in anhydrous solvents. All fabrication was carried inside an argon-filled glovebox and additional layer of spin coated PMMA was added before the device was taken out to perform measurements.

### Hall bar transport measurements

Transport measurements reported in Main Text Figs. 1–3 and Supplementary Figs. 2 and 6 were conducted in a dilution refrigerator (Bluefors) with low-temperature electronic filters and 13 T superconducting magnet. Four-terminal longitudinal resistance $R_{xx}$ and Hall resistance $R_{yx}$ were measured using standard lock-in technique with an AC excitation between 0.5 and 10 nA at around 13 Hz. Supplementary Note discusses simultaneous measurement of $R_{xx}$ and $R_{yx}$ in both domains, with measurement configuration depicted in Supplementary Fig. 1e. We present raw $R_{yx}$ and $R_{xx}$ data in Fig. 1e, f and Supplementary Fig. 2. Large $R_{yx}$ signals in both domains at low magnetic field are related to mixing with $R_{xx}$ signal due to non-ideal device geometry. This mixing does not affect topological signals at fields above 4 T which we focus on in this work. For transport measurements reported in Main Text Fig. 4 and Supplementary Fig. 7, we used physical property measurement system (PPMS, Quantum Design).

### Topological circuits measurements

To perform directional current measurements in Main Text Figs. 2–4 and Supplementary Figs. 4, 5, 7, using schematic of Fig. 2a as an example, constant current with the magnitude of 2–4 nA was injected at contact 1 and current to ground was detected with virtual-earth current preamplifiers at contacts 8 and 9. All other electrodes were floating. Supplementary Fig. 3 provides measurement details in these types of devices.

## Data availability

Source data of Figs. 1–4 can be found at: https://doi.org/10.6084/m9.figshare.20780248.v1. All other data that support the findings of this study are available from the corresponding author upon reasonable request.

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

## Acknowledgements

Authors thank Xi Wang and Jonathan Kephart for advice on device fabrication. The chiral edge current divider efforts were mainly supported as part of Programmable Quantum Materials, an Energy Frontier Research Center funded by the U.S. Department of Energy (DOE), Office of Science, Basic Energy Sciences (BES), under award DE-SC0019443. The control of the Chern numbers was mainly supported by AFOSR FA9550-21-1-0177. The authors also acknowledge the use of the facilities and instrumentation supported by NSF MRSEC DMR-1719797. J.Y. acknowledges support from the U.S. Department of Energy, Office of Science, Basic Energy Sciences, Materials Sciences and Engineering Division. C.-Z.C. acknowledges the partial support from the Gordon and Betty Moore Foundation's EPiQS Initiative (Grant GBMF9063). Y.-T.C. acknowledge support from NSF under award DMR-2004701, and the Hellman Fellowship award. X.X. and J.-H.C. acknowledge the support from the State of Washington funded Clean Energy Institute.

## Author contributions

X.X. conceived the project. D.O. and J.C. fabricated devices with assistance from Z. Lin, Z.F. and Z. Liu. D.O. and J.C. performed transport measurements with assistance from Z.F. Z. Lin performed RMCD measurements. D.X. provided theoretical support. J.Y. synthesized and characterized MnBi$_2$Te$_4$ bulk crystals. X.X., J.Y., D.X., C.-Z.C., J.-H.C., D.H.C., Y.-T.C supervised the project. All authors contribute to the data analysis. D.O., X.X., and D.H.C wrote the paper with input from all authors.

## Competing interests

The authors declare no competing interests.
