## [Peer Review File · Nature Communications]

Topological Current Divider in a Chern Insulator JunctionREVIEWER COMMENTS

Reviewer #1 (Remarks to the Author):

The manuscript reports the partitioning of chiral edge modes at the boundary of regions with different thicknesses of MnBi₂Te₄ flakes. The authors find the chiral edge channel runs along the step edge of the MnBi₂Te₄ flakes. While the formation of the chiral edge channels at the boundary between the regions with different Chern indices has been well established in the quantum Hall systems, the authors rediscovered similar physics in a different and new class of electronic system. The experimental results are new, and I agree to publish the manuscript in Nature Communications. But I think there is room for improvement in the way of presentation and recommend the authors consider the following points.

1) In quantum Hall systems of GaAs/AlGaAs heterostructure or graphene, it has been repeatedly demonstrated that chiral edge channels are formed at the boundary between the regions with different Chern numbers. In these systems, the regions with the different Chern numbers are produced by changing the carrier densities. Although the way of producing different Chern number regions is different, I think the boundary chiral edge channels in quantum Hall systems and the present system possess a common underlying origin. Therefore, I recommend the authors mention these studies by citing proper papers.

2) It is not clear to me why the 6-nm-thick MnBi₂Te₄ is the C=2 Chern insulator accommodating two chiral edge modes. I understand the mechanism for the formation of the C=2 Chern insulator is not the central topic of the present study. However, because the thickness dependence of the Chern number is at the heart of the present study, a brief explanation or a summary for the current understanding of the Chern numbers in MnBi₂Te₄ would help the readers to understand the authors' findings.

3) Please present a photo image of Device 1. Because the electrical measurements presented in the manuscript are not conventional ones, the resistance or the current values are supposed to be sensitive to the actual device dimensions and the choice of the electrodes. Although a photo image of Device 1 is presented in the supplemental materials, apparently it does not correspond to the schematic in Fig. 2a. Put the numbers of the electrodes on the photo image so that the correspondence becomes clear.

4) The authors do not specify the electrodes used for the measurement of R_{xx} and R_{yx} in Fig. 1 and Supplementary Fig. 2. I guess the authors sent the current across the step edge (from the left end to the right end in Fig. 1c). In such a case, the injected current is partially reflected due to the chiral edge mode along the step edge. The residual surface bulk conduction would make the current density distribution more complicated. Under such a complicated current distribution, is it possible to separate the signals of Domain I from that of Domain II as in Fig. 1e and 1f? In my opinion, the data in Fig. 1e and Fig. 1f are already affected by the presence of the boundary i.e. the presence of the other domain, therefore the data are not necessarily identical to the Hall resistance of individual domains. At least an explanation would be required for how R_{xx} and R_{yx} are measured and why one set of the data belongs to Domain I and the other to Domain II.

5) The authors do not explain the required characters of the step edge to accommodate the boundary chiral edge mode. Is there the minimum or the maximum height difference to form the chiral edge mode? Is the atomic arrangement of the step edge relevant to the chiral edge mode? What is the success rate of the formation of the chiral edge mode when a step edge is formed in a sample?

6) On page 4, the authors describe "More generally, the topological nature of the device makes the use of different contacts within a single topological domain equivalent". I disagree with this statement. The equivalence is guaranteed by the vanishingly small surface bulk conductivity $\sigma_{xx} \sim 0$. Hence, this is a rather special situation, not in general situations.

7) In the discussion part, the authors mention the high-frequency nonreciprocal switches. I wonder how high frequency can be reached by using MnBi₂Te₄ and what material parameter sets the high-frequency cut-off.

8) In Fig. 3a, the authors present the gate voltage dependence of the current measured using various pairs of drain electrodes. Besides the plateau region, the gate voltage dependences look similar to each other at $V_{bg} < 85$ V despite the use of different electrodes. Why they are similar to each other?

Curiously, in the gate voltage range $95 \text{ V} < V_{\text{bg}} < 110 \text{ V}$, the values of I_9 are smaller than 50 % in the upper two panels and larger than 50 % in the lower two panels. Why does such a difference appear in the particular voltage range and what is the qualitative difference between the electrodes 4, 5 and 6, 7?

Reviewer #2 (Remarks to the Author):

The manuscript by Ovchinnikov et al. reported the realization of “Topological Current Divider in a Chern Insulator Junction” in 2D MnBi₂Te₄ samples. This is a very clearly written paper with high quality data. The data clearly supports the conclusion.

I have one minor suggestion: the authors should consider to show also antisymmetrized R_{yx} data in Figs. 1e,f. Currently, the data show also very large R_{yx} at zero magnetic field but I assume that those R_{yx} signals are not real Hall signals.

Reviewer #3 (Remarks to the Author):

In this manuscript, the authors used a series of magneto-transport measurements to investigate topological domains with different Chern numbers in MnBi₂Te₄ and chiral edge current at the boundary between these domains. They demonstrated that chiral edge current at a step in thickness of an exfoliated MnBi₂Te₄ flakes can be controlled by combination of gate voltage and magnetic field. The results are convincing. Figure 2 shows the most striking effect, where the electrical current injected at the edge can be either split into 1:1 or completely directed through certain path, depending on the direction of the chiral edge current circulation set by the sign of the magnetic field (+12 T vs -12 T). The authors further show in Figure 3 that the divided current remains the same for different contacts used in domain I, which is consistent with a dissipationless edge current. Considering that dissipationless chiral edge transport is of high interest, and MnBi₂Te₄ is currently the best material/platform to study this physics, I recommend this paper to be accepted for publication after addressing some questions and concerns:

1. It is not entirely clear in Figure 1 how R_{yx} is measured, since there are two domains on the sample. For example, does the excitation current flow through both domains or just a single one? There are no electrode diagrams in the figure. One solution would be to show an optical image of the device and specify which electrodes are used for the measurement. Another solution would be to include electrode diagrams as insets to Figures 1e and 1f.

2. Figures 1c and 1d show two possible edge current configurations for the $C=1 + C=2$ state. Can the transport measurements distinguish between the two, and if so, then which one is realized? There appears nothing written about this in the main text. I only see some discussion of this point in the caption of supplementary figure 4. I would strongly recommend that this discussion is included in the main text.

3. In ref. 28 of the manuscript, the MnBi₂Te₄ enters $C=2$ regime when the thickness ≥ 8 layers, while in this manuscript the MnBi₂Te₄ enters $C=2$ regime with thickness of 6 layers. What is the reason for this difference?

4. What does the black color mean in the Supplementary Fig 2a.?

5. In Figure 3a, when the contact 4 and 5 are used, the back gate voltage range for current divider is smaller than the case when the contact 6 and 7 are used. What's the reason for this?

6. in Figure 2b and 3a, outside the quantized region (roughly $85V < V_{bg} < 95V$, in yellow), the currents are no longer split 50:50. Sometimes the current through the domain I electrode is higher than the domain II electrode, and sometimes it is lower. Is this understood in terms of bulk and edge resistivities?

We sincerely thank all three referees for their thoughtful and constructive comments. In the following, we provide a point-to-point response to their comments. Major changes are shown in red in the revised manuscript.

Reviewer #1:

The manuscript reports the partitioning of chiral edge modes at the boundary of regions with different thicknesses of MnBi₂Te₄ flakes. The authors find the chiral edge channel runs along the step edge of the MnBi₂Te₄ flakes. While the formation of the chiral edge channels at the boundary between the regions with different Chern indices has been well established in the quantum Hall systems, the authors rediscovered similar physics in a different and new class of electronic system. The experimental results are new, and I agree to publish the manuscript in Nature Communications.

Response: We appreciate that the reviewer finds our results interesting and recommends our work for publication in Nature Communications.

But I think there is room for improvement in the way of presentation and recommend the authors consider the following points.

1) In quantum Hall systems of GaAs/AlGaAs heterostructure or graphene, it has been repeatedly demonstrated that chiral edge channels are formed at the boundary between the regions with different Chern numbers. In these systems, the regions with the different Chern numbers are produced by changing the carrier densities. Although the way of producing different Chern number regions is different, I think the boundary chiral edge channels in quantum Hall systems and the present system possess a common underlying origin. Therefore, I recommend the authors mention these studies by citing proper papers.

Response: Thank you for this suggestion, which will hopefully allow us to present a more complete set of references and to highlight the novelty of our work. We now updated the second paragraph of Introduction part of the main text (page 2):

This was demonstrated in experiments on two-dimensional electron gas¹¹ and graphene¹²⁻¹⁴ where regions with different C can be created by means of locally changing carrier densities in quantum Hall regime. The Chern insulator networks¹⁵⁻¹⁷ comprised of domains with distinct Chern numbers may lead to complex device architectures, where C of individual domains will control topological current properties, such as current amplitude and propagation direction. Previous attempts to construct such networks which went beyond quantum Hall state relied on chiral edge states existing at domain walls in magnetic topological insulators, since domains of opposite magnetization have opposite Chern number (Ref.¹⁸⁻²⁰).

2) It is not clear to me why the 6-nm-thick MnBi₂Te₄ is the $C=2$ Chern insulator accommodating two chiral edge modes. I understand the mechanism for the formation of the $C=2$ Chern insulator is not the central topic of the present study. However, because the thickness dependence of the Chern number is at the heart of the present study, a brief explanation or a summary for the current understanding of the Chern numbers in MnBi₂Te₄ would help the readers to understand the authors' findings.

Response: Our recent work (Cai et al., Nature Communications 13, 1668 (2022), also Ref. 35 in our manuscript) provides a detailed discussion of the $C = 2$ formation. Briefly, $C = 2$ is composed of $C = 1$ Chern insulator state and $\nu = 1$ quantum Hall state. Regarding thickness dependence, we observe $C = 2$ in flakes which are ≥ 6 -SL thick. To be clearer on this point, we now write in the first paragraph of the Chern insulator junction formation part of the main text (pages 2-3):

Chern number $C = 2$ state in MnBi_2Te_4 was recently identified as a combination of $C = 1$, which originates from non-trivial band topology and $\nu = 1$ quantum Hall state³⁵. Within our experimental parameter space^{32,35}, we observe $C = 2$ only in flakes which are ≥ 6 layers thick (see Methods). In flakes with thickness 4 and 5 layers only $C = 1$ state appears.

3) Please present a photo image of Device 1. Because the electrical measurements presented in the manuscript are not conventional ones, the resistance or the current values are supposed to be sensitive to the actual device dimensions and the choice of the electrodes. Although a photo image of Device 1 is presented in the supplemental materials, apparently it does not correspond to the schematic in Fig. 2a. Put the numbers of the electrodes on the photo image so that the correspondence becomes clear.

Response: We now provide image of the device with electrodes assignment in modified Supplementary Figure 1 (panels d and e).

4) The authors do not specify the electrodes used for the measurement of R_{xx} and R_{yx} in Fig. 1 and Supplementary Fig. 2. I guess the authors sent the current across the step edge (from the left end to the right end in Fig. 1c). In such a case, the injected current is partially reflected due to the chiral edge mode along the step edge. The residual surface bulk conduction would make the current density distribution more complicated. Under such a complicated current distribution, is it possible to separate the signals of Domain I from that of Domain II as in Fig. 1e and 1f? In my opinion, the data in Fig. 1e and Fig. 1f are already affected by the presence of the boundary i.e. the presence of the other domain, therefore the data are not necessarily identical to the Hall resistance of individual domains. At least an explanation would be required for how R_{xx} and R_{yx} are measured and why one set of the data belongs to Domain I and the other to Domain II.

Response: We agree with the reviewer that in general case care should be taken in interpreting measurements where the current is passed through domains with different properties. However, the presence of edge mode on the boundary should not affect the determination of Chern numbers in our device geometry.

To illustrate that, in Figure R1 we show a measurement schematic (now also included in Supplementary Figure 1e) and possible edge states configurations.

- In diffusive transport regime (panel b), the feasibility of R_{xx} and R_{yx} measurement for each domain can be proved by using the theorem of equivalent circuits: when measuring Domain I (or Domain II), the other domain could be treated as the source (or drain) contact resistance. The same equivalence is still applicable when one domain is in topological transport regime and in the other one transport is diffusive.

- When the transport is purely topological (panels c-d), both domains would have insulating bulk and the transport happens on the edge. Now, the transport is described by Landauer formula $I_i = \frac{e^2}{h} \sum_j (T_{j,i} V_j - T_{i,j} V_i)$, where $T_{i,j} = N$ when N edge states connect terminal i to terminal j and $V_i(I_i)$ denotes the voltage (current) on terminal i . This leads to a connected current network that ensures the applicability of the above equivalence. To demonstrate it for the case of $C_I = 1$ and $C_{II} = 2$, we have set $I_1 = -I_6 = I_s$ and Landauer formula yields: $V_2 = V_3 = -\frac{1}{2} \times \frac{h}{e^2} I_s$, $V_4 = V_5 = V_6 = -1 \times \frac{h}{e^2} I_s$, $V_7 = V_8 = V_9 = V_{10} = V_{11} = V_1 = 0$, corresponding to our measurement: $R_{xx}^I = \frac{V_4 - V_5}{I_s} = 0$, $R_{xx}^{II} = \frac{V_2 - V_3}{I_s} = 0$ and $R_{yx}^I = \frac{V_8 - V_5}{I_s} = \frac{h}{e^2}$, $R_{xx}^{II} = \frac{V_{11} - V_2}{I_s} = \frac{1}{2} \times \frac{h}{e^2}$. Transport measurements with geometry like ours in domain-type of devices have been performed in multiple publications, see for example Yasuda et al., Science 358, 1311–1314 (2017) and Rosen et al., NPJ Quantum Mater. 2, 1–6 (2017).

Figure R1 | Transport measurement schematic and different Chern numbers. a, Transport measurement schematic for determination of Chern numbers of each individual domain. **b-d,** Chern numbers of each domain and corresponding propagation of edge modes.

We add this discussion in Supplementary Note (page 2 of the Supplementary Information).

5) The authors do not explain the required characters of the step edge to accommodate the boundary chiral edge mode. Is there the minimum or the maximum height difference to form the chiral edge mode?

Response: The key to our observation is that neighboring domains have different Chern numbers ($C = 1$ and $C = 2$). In our case, we achieve that by choosing a flake that has two regions with different thickness. We have studied devices with thickness difference of 1 (Device #2) and 2 (Device #1) septuple layers. We do not know for sure the allowed maximum height difference. However, based on our experience, the minimal thickness for $C = 1$ state is 4 septuple layers and the maximum thickness for the quantized devices is 8 layers. We speculate that the maximum height difference could be 4 septuple layers.

Is the atomic arrangement of the step edge relevant to the chiral edge mode?

Response: Chiral edge mode which we discovered on the interface between two domains with different Chern numbers in MnBi_2Te_4 is identical to edge modes in quantum anomalous Hall (QAH) and quantum Hall (QH) systems. Such edge modes at a given direction of winding are immune to local disorder and exist regardless of local atomic arrangements.

What is the success rate of the formation of the chiral edge mode when a step edge is formed in a sample?

Response: Whenever the condition of $C_I = 1$ and $C_{II} = 2$ is met, as it is in two devices in this study, we observe a chiral edge state at the step edge between two domains.

6) On page 4, the authors describe “More generally, the topological nature of the device makes the use of different contacts within a single topological domain equivalent”. I disagree with this statement. The equivalence is guaranteed by the vanishingly small surface bulk conductivity $\sigma_{xx} \sim 0$. Hence, this is a rather special situation, not in general situations.

Response: We thank the reviewer for bringing up this point, we agree that our claim lacks clarity. We now changed this sentence to make our statement more precise:

The existence of chiral edge states in this device makes the use of different contacts within a single topological domain equivalent^{6,7}.

7) In the discussion part, the authors mention the high-frequency nonreciprocal switches. I wonder how high frequency can be reached by using MnBi_2Te_4 and what material parameter sets the high-frequency cut-off.

Response: Disks fabricated from 2DEG in quantum Hall regime and magnetic topological insulators have been demonstrated to host edge plasmons (please see Andrei et al., Surf. Sci. 196, 501 (1988), Kumada et al., Phys. Rev. B 84, 045314 (2011), Mahoney et al., Phys. Rev. X 7, 011007 (2017), Mahoney et al., Nat. Commun. 8, 1836 (2017)). In analogy to non-reciprocity, demonstrated in DC response of chiral edge modes, those edge plasmons also exhibit non-reciprocity, which allows to engineer of high-frequency non-reciprocal passive elements such as circulators, which operate around the frequencies of 1-2 GHz.

MBT has many similarities to the described QH and QAH systems, so we anticipate that similar fundamental physics of edge plasmons would be at play. This should in turn enable high-frequency non-reciprocal operation. At the same time, some advantages of MBT in comparison abovementioned materials systems include higher operational temperature, as we discuss in Figure 4. We hope that our work will draw attention to MnBi_2Te_4 as a possible platform for such devices and that fundamental studies of microwave response will quickly emerge. At this stage, exact parameters which will determine the cut-off frequency and other device parameters are not known and will require studies of edge plasmons properties in MBT, which is beyond the scope of this manuscript.

We now added references to the sentence referring to high-frequency operation:

Finally, although our results are in the low-frequency limit, the device architecture offers a promising route to higher frequency non-reciprocal switches^{6,7} based on van der Waals topological magnets.

8) In Fig. 3a, the authors present the gate voltage dependence of the current measured using various pairs of drain electrodes. Besides the plateau region, the gate voltage dependences look similar to each other at $V_{bg} < 85$ V despite the use of different electrodes. Why they are similar to each other? Curiously, in the gate voltage range $95 \text{ V} < V_{bg} < 110$ V, the values of I_9 are smaller than 50 % in the upper two panels and larger than 50 % in the lower two panels. Why does such a difference appear in the particular voltage range and what is the qualitative difference between the electrodes 4, 5, and 6, 7?

Response: This is an excellent question. The challenge of understanding this within the V_{bg} regime suggested by the reviewer ($95 \text{ V} < V_{bg} < 110$ V) is that in Domain II there is an incipient $C = 3$ state at this experimental condition. We note that we have already observed $C = 3$ state in other high-quality MBT devices (see Cai et al., Nature Communications 13, 1668 (2022)). In this device, $C = 3$ state is not fully developed, possibly due to lower mobility in PMMA-coated samples than in BN-covered samples which are mainly discussed in the paper above. On the other hand in Domain I, $C = 1$ state starts to fade out with R_{yx} going away from h/e^2 value as we increase V_{bg} . This results in a complex situation when measurable edge conductance coexists with increasing bulk conductance as we increase V_{bg} .

Based on that we can speculate that electrodes 4 and 5 are located close enough to the junction so that more current gets injected into Domain I from edge states in Domain II running clockwise. As we describe above transport in Domain I is a combination of measurable edge and bulk transports which implies that less current will get to electrodes located further away from the junction region through edge states. As soon as electrodes 6, 7, and 8 are further from the junction less current is being injected there.

Reviewer #2:

The manuscript by Ovchinnikov et al. reported the realization of “Topological Current Divider in a Chern Insulator Junction” in 2D MnBi₂Te₄ samples. This is a very clearly written paper with high quality data. The data clearly supports the conclusion.

Response: We sincerely thank the reviewer for complementing our manuscript and data quality.

I have one minor suggestion: the authors should consider to show also antisymmetrized R_{yx} data in Figs. 1ef. Currently, the data show also very large R_{yx} at zero magnetic field but I assume that those R_{yx} signals are not real Hall signals.

Response: Reviewer is correct that the origin of high R_{yx} in the low magnetic field region ($-4 \text{ T} < \mu_0 H < 0 \text{ T}$) near the charge neutrality point is indeed related to R_{xx} contribution due to non-perfect device geometry. To perform proper antisymmetrization of R_{yx} signals in samples with topological signals and magnetism (see for example Deng et al., Science 367, 895–900 (2020)), ideally maps

similar to Figure 1e, f should be recorded with magnetic field swept both up and down. In our case only sweep down magnetic field-gate map was recorded. However, R_{xx} contribution to R_{yx} does not affect the sector of phase diagram where we observed edge transport and demonstrate chiral edge current division. Thus, we prefer to present the raw data. To clarify that we now write in the Methods section:

We present raw R_{yx} and R_{xx} data in Figures 1e, f and Supplementary Figure 2. Large R_{yx} signals in both domains at low magnetic field are related to mixing with R_{xx} signal due to non-ideal device geometry. This mixing does not affect measurements at fields above 4 T which we focus on in this work.

Reviewer #3:

In this manuscript, the authors used a series of magneto-transport measurements to investigate topological domains with different Chern numbers in MnBi₂Te₄ and chiral edge current at the boundary between these domains. They demonstrated that chiral edge current at a step in thickness of an exfoliated MnBi₂Te₄ flakes can be controlled by combination of gate voltage and magnetic field. The results are convincing. Figure 2 shows the most striking effect, where the electrical current injected at the edge can be either split into 1:1 or completely directed through certain path, depending on the direction of the chiral edge current circulation set by the sign of the magnetic field (+12 T vs -12 T). The authors further show in Figure 3 that the divided current remains the same for different contacts used in domain I, which is consistent with a dissipationless edge current. Considering that dissipationless chiral edge transport is of high interest, and MnBi₂Te₄ is currently the best material/platform to study this physics, I recommend this paper to be accepted for publication after addressing some questions and concerns:

Response: We greatly appreciate the reviewer's high opinion about our work and for the recommendation to publication of our work in Nature Communications.

1. It is not entirely clear in Figure 1 how R_{yx} is measured, since there are two domains on the sample. For example, does the excitation current flow through both domains or just a single one? There are no electrode diagrams in the figure. One solution would be to show an optical image of the device and specify which electrodes are used for the measurement. Another solution would be to include electrode diagrams as insets to Figures 1e and 1f.

Response: We have clarified the measurement geometry based on reviewer's suggestion. Please see the updated Supplementary Figure 1, panels d and e for device optical micrograph and electrodes used for each transport measurement.

2. Figures 1c and 1d show two possible edge current configurations for the $C=1 + C=2$ state. Can the transport measurements distinguish between the two, and if so, then which one is realized? There appears nothing written about this in the main text. I only see some discussion of this point in the caption of supplementary figure 4. I would strongly recommend that this discussion is included in the main text.

Response: Thank you for this question. When our samples are tuned to non-zero Chern numbers by the gate voltage, magnetic field transport measurements can indeed distinguish one configuration of edge states from the other. The key differences between the two cases depicted in Figure 1c and Figure 1d are the following:

- 1) There is continuity of edge mode going from Domain II to Domain I in Figure 1c.
- 2) Two regions with different Chern numbers are decoupled in Figure 1d, in other words, they do not share a common edge mode.
- 3) There is a single edge mode going along the junction in Figure 1c.
- 4) There are three edge modes (two going up and one going down) going along the junction in Figure 1d.

Our measurements in Figures 2, 3, and Supplementary Figure 4 allow us to determine the exact configuration of edge modes as discussed below:

- a) Figure 1a and b allow us to determine that there is a shared chiral edge mode for two domains, which confirms point 1) from the list above. In case the configuration depicted in Figure 1d was correct, all current would flow to terminal 9 and none would reach terminal 8.
- b) Supplementary Figure 4 confirms that there is only a single edge mode on the junction, which confirms point 3) from the list above. In case the configuration depicted in Figure 1d was correct most of the current would flow to terminal 8.

To be clear that our experiments directly identify the configuration of chiral edge modes, we now added the following paragraph to the Topological Current Divider part of the main text (page 3):

This measurement unambiguously identifies that Fig. 1c depicts the correct interfacial chiral edge conduction channel configuration. In case the physical picture depicted in Fig. 1d would be a valid one, all current should flow from contact 1 to 9 and give vanishing I_8 . The configuration in Fig. 1c leads to the current flow pattern sketched using white lines in Fig. 2a, which leads to $I_9 = I_8$ (see Methods and Supplementary Figure 3 for measurement details).

3. In ref. 28 of the manuscript, the MnBi₂Te₄ enters C=2 regime when the thickness ≥ 8 layers, while in this manuscript the MnBi₂Te₄ enters C=2 regime with thickness of 6 layers. What is the reason for this difference?

Response: As the reviewer mentioned, current literature provides inconsistent results regarding thickness dependence of $C = 2$ observations. We found through systematic studies that $C = 2$ states can be observed in samples which are between 6-SL and 8-SL thick layers thick (with our high single crystal quality and fabrication). This has been reported in detail in Ref. 35 (Cai et al., Nature Communications 13, 1668 (2022)). Further advances in both bulk crystals growth and device fabrication will allow us to better understand thickness-dependent properties. In our view, fewer defects and more careful fabrications may lead to higher mobility and observation of $C = 2$ state in thinner samples.

4. What does the black color mean in the Supplementary Fig 2a.?

Response: Our measurements are done with the lock in amplifier. Due to the transition from $C = 1$ to $C = 0$ at fields below -4 T and 50 mK measurement temperature, this particular sample enters very high resistance state where detection with lock in amplifier becomes challenging. Thus, our setup does not allow a good measurement of the resistive state of the 4-SL sample at the low field limit. We emphasize that the absence of measurable R_{xx} in that region of the phase diagram does not affect any conclusions of our present work.

We now added the following to the caption of Supplementary Figure 2a:

In the region $80 \text{ V} < V_{bg} < 85 \text{ V}$ and $-4 \text{ T} < \mu_0 H < 0 \text{ T}$, Domain I becomes too resistive for lock-in type measurements (area marked with black color).

5. In Figure 3a, when the contact 4 and 5 are used, the back gate voltage range for current divider is smaller than the case when the contact 6 and 7 are used. What's the reason for this?

Response: We anticipate small inhomogeneity in Fermi level location across the sample. It can be related to disorder originating from PMMA, which covers the top surface and may host charge puddles, as well as to the inhomogeneous distribution of defects, which may cause Fermi level variation across the device (see for example J.-Q. Yan 2022 ECS J. Solid State Sci. Technol. 11 063007). This may lead to a variation of $C = 1$ or $C = 2$ states as a function of V_{bg} and small variations in the voltage range at which interfacial edge state between two domains is being formed. In addition, devices presented in this study have large lateral dimensions for exfoliated flakes ($\sim 100 \mu\text{m}$) which makes inhomogeneity of Fermi level within the device plausible.

We now added the following explanation on page 4 of the main text:

Note that small changes in the gate range for current divider operation in Fig. 3a might be related to small Fermi level variations across the device, which has a large lateral size of $\sim 100 \mu\text{m}$.

6. in Figure 2b and 3a, outside the quantized region (roughly $85\text{V} < V_{bg} < 95\text{V}$, in yellow), the currents are no longer split 50:50. Sometimes the current through the domain I electrode is higher than the domain II electrode, and sometimes it is lower. Is this understood in terms of bulk and edge resistivities?

Response: Thank you for this question. As an example, we can have a look at the device schematic in Figure 2a and track the current getting to terminals 8 and 9 in Figure 2b at $\mu_0 H = -12$ T. We can split the gate range into several regions:

- 1) For $20 \text{ V} < V_{bg} < 70 \text{ V}$ both domains are in diffusive regime of transport. Terminal 9 is closer to terminal 1 than terminal 8. Thus, the majority of the current will flow from terminal 1 to terminal 9 through the diffusive bulk.
- 2) For $75 \text{ V} < V_{bg} < 85 \text{ V}$ peak in current flowing to terminal 8 is observed. Change of V_{bg} in each Domain leads to $C_I = 1$ and $C_{II} = 1$ formation. However, this formation does not happen simultaneously, leading to a complex situation when emerging topological edge modes and residual bulk conduction contribute to ratio between currents that reach terminals 8 and 9.

- 3) Further increasing V_{bg} to 85 V leads to $C_{II} = 2$ formation, edge state forms at Chern junction, both currents become identical at $85V < V_{bg} < 95V$.
- 4) For $95 V < V_{bg}$ please see below the reply to Reviewer #1, question 8.

The challenge of understanding this within the V_{bg} regime suggested by the reviewer ($95 V < V_{bg} < 110 V$) is that in Domain II there is an incipient $C = 3$ state at this experimental condition. We note that we have already observed $C = 3$ state in other high-quality MBT devices (see Cai et al., Nature Communications 13, 1668 (2022)). In this device, $C = 3$ state is not fully developed, possibly due to lower mobility in PMMA-coated samples than in BN-covered samples which are mainly discussed in the paper above. On the other hand in Domain I, $C = 1$ state starts to fade out with R_{yx} going away from h/e^2 value as we increase V_{bg} . This results in a complex situation when measurable edge conductance coexists with increasing bulk conductance as we increase V_{bg} .

Based on that we can speculate that electrodes 4 and 5 are located close enough to the junction so that more current gets injected into Domain I from edge states in Domain II running clockwise. As we describe above transport in Domain I is a combination of measurable edge and bulk transports which implies that less current will get to electrodes located further away from the junction region through edge states. As soon as electrodes 6, 7, and 8 are further from the junction less current is being injected there.

REVIEWER COMMENTS

Reviewer #1 (Remarks to the Author):

The authors have addressed all my comments satisfactorily. I think the manuscript is now ready for publication.

Reviewer #3 (Remarks to the Author):

The authors have provided very good responses to my concerns and updated the manuscript and supplemental accordingly. The additional details they provide have substantially clarified the text. I strongly recommend publication of this work in Nature Communications.

We sincerely thank all referees for their thoughtful and constructive comments. In the following, we provide a point-to-point response to their comments.

Reviewer #1:

The authors have addressed all my comments satisfactorily. I think the manuscript is now ready for publication.

We are glad that we managed to address all comments of the reviewer. We appreciate that the reviewer recommends our work for publication in Nature Communications.

Reviewer #3:

The authors have provided very good responses to my concerns and updated the manuscript and supplemental accordingly. The additional details they provide have substantially clarified the text. I strongly recommend publication of this work in Nature Communications.

We are glad that we were able to address all concerns of the reviewer and update the manuscript accordingly. We thank the reviewer for positively assessing our work and for the recommendation of our work for publication in Nature Communications.